# Construction Sector Role in Gross Fixed Capital Formation: Empirical Data from Russia

**Elena Stupnikova and Tatyana Sukhadolets ***

Department of Economics, Russian University of Transport (MIIT), 127994 Moscow, Russia;
stupnikovaea@yandex.ru
*   Correspondence: sutv@yandex.ru

**Abstract:** The purpose of this study was to research and understand the interrelations between the growth of gross fixed capital formation (GFCF), the volume of construction industry, supply of interindustry balance, and amount of fixed-asset investments in Russia between 2000 and 2016. The autoregressive distributed-lagged (ARDL) bound testing methodology and regression analysis were applied to evaluate the cointegration and influence of construction industry volume on gross fixed-capital formation. Empirical studies on the role of the construction industry are at the forefront of economic research; however, ARDL modeling studies of GFCF have yet to be conducted in Russia. The study revealed a non-linear causation between construction industry volume and the growth in GFCF over a long time period. The correlation was stationary and cointegrated. Fixed investment positively affected gross fixed capital formation only in periods of economic expansion, whereas the effectiveness of fixed-asset investments had greater volatility in times of crisis. The construction industry was not practically affected by crisis shocks, demonstrating a permanent stationarity in the causal relationship with GFCF, whereas causal relations between GFCF and the supply of interindustry balance were absent. The results are important for further research in the field of economic growth, the development of a national budget and investment policy, as well as investment project selection.

**Keywords:** gross fixed capital formation; construction; fixed-asset investments; non-linear ARDL

## 1. Introduction

Since the early 2000s, the Russian economy has grown considerably. During the period between 2000 and 2008, the growth in gross domestic product (GDP) was 21%–23% per year on average; however, in 2009, the growth dynamics within the year were negative (–1%) (Goryunov et al. 2015). Other observations include the downturn in 2013 when GDP contracted by 7.29%; in 2015, it contracted by 1.07%; and in 2018, by 1.34% (Kudrin and Gurvich 2014). Since 2000, various national reforms (social, infrastructural, military, agricultural, management, etc.) have been implemented in Russia (Kudrin and Sokolov 2017).

One of the major purposes of this study was to identify the role and measure the influence of the construction industry on the gross fixed capital formation (GFCF), as the construction industry in Russia is developing with the strong financial support of the government. In national accounts, GFCF is applied to measure the value of fixed assets excluding amortized costs (Sarmento 2018) and represents the increase in the tangible and intangible assets produced. Statistically, the GFCF measures the value of new or existing fixed asset acquisitions of government, business, and private households. Infrastructure construction contributes to altering the GFCF (Rauf et al. 2018; Okoye et al. 2017).

In connection with Russian economic reforms, the GFCF most reflects the extent of the accumulates process of economic changes, as, in the past 15 year since the mid-2000s, the government has financed many projects from various sectors of the economy: Construction, healthcare, science, the social sector,

etc. GFCF is defined as the acquisition of new or second-hand assets and thus shows something about how much of the new value added in the economy is invested rather than consumed. This article attempts to answer several questions: (1) What is the role of the construction industry in changes to the GFCF? (2) Do consistent patterns exist there? What are their specifications? (3) Does the financial support of these sectors provided by the government contribute to economic growth?

Given the absence of such studies in Russia, it was necessary to accomplish this research and clarify the causal relationship between the construction industry and GFCF with the autoregressive distributed-lagged (ARDL) bound testing methodology, especially for crisis periods.

The development of the economy of different countries has been characterized using factors that facilitate the investing process in the field of construction as the instrument of structural variation in fixed capital formation (Liu et al. 2016). The facilitative tendencies of financing construction investment persist and develop. Conditions, which prevent the effective transformation of liquid capital into elements of real infrastructure capital, are formed to stimulate the investment process in the real economy. State investments in the construction industry have significantly furthered infrastructure development in Russia. Many studies on the role of the construction industry in regional economic growth have been published, which corroborate their close relationship (Silka 2017; Jonga et al. 2008; Okoye et al. 2017). The economic growth literature demonstrates a non-linear correlation between the amount of state financing and economic growth. This correlation is a U-shaped curve, which assists with determining the optimal proportion of public spending (Mohammad and Falahib 2018). During the economic turbulences, recessions, or deep crises, actors prefer to hold liquidity (Sahin 2018). The roles of public and private investments, economic growth, and the growth of the construction sector were observed by (Bahal et al. 2018). Fluctuations in the GFCF predict future business activity, business confidence, and the tendency of economic growth (Cohen et al. 2012). Each dollar spent on infrastructure construction is distributed among workers, equipment suppliers, intermediaries, and among other related fields; thus, the entire economy is indirectly affected (Cohen et al. 2012). Toward the middle of the 2000s, the Russian economy was opened to the free movement of short-term investments, overtaking China, India, and Brazil in the degree of openness of financial markets for cross-border speculative operations. In developed countries, a minimum amount of investment in the construction industry has been established, which depends on the level of gross national product (GNP), which is measured considering the minimum level of additional value required for the growth of GNP (Lopes et al. 2002). There are arguments that explain the dependence between the condition of the investment climate in the private sector (including the construction industry) and the inflow–outflow movements of assets (Konopczak and Konopczak 2017). For Turkey, Sahin (Sahin and Berument 2019) explored the effects of Central Bank (C.B.F.) of the Republic of Turkey on banking sector credits by using the nonlinear autoregressive distributed lag model (N.A.R.D.L.), which also allows for the asymmetric response of loans to changes in C.B.F. The model also assesses this effect for short- and long-term asymmetry.

The Russian economy is dependent on the revenue from the oil and gas industry. Analytical studies on oil rent, which is colloquially known as "resource cures", have demonstrated the dependence of gross domestic product (GDP) growth on oil rent (Neumayer 2004a). GDP calculation includes the natural and other amortization of capital in the form of income; however, from the viewpoint of real income, the growth in a resource-intensive economy is less significant than the growth in the GDP. To some extent, resource cures are associated with unsustainable over-consumption. The oil rent in Russia, which is 40%–50% of its budgeted revenues, is primarily spent on the consumption (Simola and Solanko 2017). Even the reserve fund and the national welfare fund savings are considered as reserves by the government for supporting the current consumption level in cases of adverse changes in market conditions and the outcomes of sanctions, instead of investing in the national economy (Simola and Solanko 2017). Problems that slow the Russian economy are chronic and cannot be solved with simple solutions. In Russia, one of the major reasons for the market environment weakness and slow economic growth is the dominant participation of the state and quasi-public

companies (Kudrin and Gurvich 2014). Therefore, from the viewpoint of GFCF variation, the influence of profound state participation in infrastructure construction projects is a matter of interest. There have been large-scale studies on the role of the construction sector in GDP (Turin 1973; Wells 1987; The World Bank 1984). Studies since the 2000s, based on macroeconomic analysis, have attempted to simulate the relationship between the construction industry and economic growth (Lopes 1998; Ozkana et al. 2012; Tardieua et al. 2015). In developing countries, the quality and accessibility of such data are issues. Paradigms (based on Keynesian theory) have emerged that represent the construction industry as an agent that supports the growth in the economy at different phases of development (Liu et al. 2016; Lopes et al. 2002; Tardieua et al. 2015).

The remainder of this article is structured as follows: The theoretical and empirical literature is described in Section 2; Section 3 includes the methodology and related data; Section 4 provides an analysis of the empirical results; and a discussion of the data is provided in Section 5.

## 2. Literature Review

In most cases, to evaluate the economic development of a country, GDP and GNP are used (Qin et al. 2006). However, GFCF measures the net growth of the fixed capital in dynamics. The approach based on the changing economic development and dynamic of GFCF is infrequently applied (Bazilian et al. 2011). Few studies have analyzed any form of connection between construction and GFCF (Gruneberg and Fraser 2013), although GFCF statistically measures the value of acquisitions of new or existing fixed assets. This measurement connects GFCF with construction, since it is construction that participates in the creation of new assets. Many studies have been devoted to the study of GFCF and macroeconomics and economic growth (Bleaney and Greenaway 2001; Holz 2006; Apergisa and Payneb 2010; Qin et al. 2006; Mau 2017; Bahal et al. 2018) and many reports have been published on construction and investment in construction (Lopes 1998; Amiril et al. 2014; Ozkana et al. 2012; Okoye et al. 2017). Aggregation of GFCF through the assessment of fixed assets (excluding amortized costs) is corrected by the system of national accounts (SNA). This system is the international standard that was developed for countries of the European Union (SNA 2018). Russian SNA is at the inception development stage. Particularly, problems have arisen due to the lack of data on capital account operations because it is impossible to form indicators of savings and capital transfers. Gross accumulation, which is an element of GNP, demonstrates changes in the value of the produced non-financial assets and reflects the results of the investment activities of the region. Russia's gross fixed capital formation data remain in active status but have been delayed for two years. Despite the difficulties arising from the Russian GFCF statistics, evaluating procedures have been integrated; thus, the research data are sufficient to analyze the GFCF changes in evaluation assessment. A similar situation with GFCF is observed in many countries, such as China (Holz 2006). The procedures of the GFCF assessment methodologies in Russia have several limitations: (1) Gross fixed capital formation is not equal to the investment amount; (2) the value of fixed capital, which is newly created by investments, is not equal to these investments; (3) amortized expenses are excluded; and (4) official data are included with significant changes of the revaluation conducted in the 2000s. Therefore, assessment procedures can be accomplished with the available Russian public statistics data.

Assessments of the factors influencing economic growth and gross capital formation are associated with various regularities: The increase in trade volume (Barro 1991; Baek and Yang 2010; Bleaney and Greenaway 2001), changes in the costs of non-ferrous metals (Mardonesa and Riob 2019), financial development and mediation (Sevena and Yetkinerb 2016), and technological innovations (Mau 2017). A long-term balance exists between real GDP, real gross fixed capital formation, and labor force; this balance is positive and statically significant (Apergisa and Payneb 2010). The political environment influences gross capital formation, especially after current Western sanctions were imposed. The sanctions not only affect state-controlled banks and gas and oil companies but have restricted the ability to obtain external loans due to the limitations on foreign investments (Gurvich and Prilepskiy 2015). In the period of 2014 to 2017, Russian gross capital flows decreased by 280 billion rubles. The estimated

impact of sanctions on GDP is significant (−2.4 percentage points in 2017). However, this is 3.3 times lower than the expected influence of the oil price (Gurvich and Prilepskiy 2015).

The previously mentioned factors may not improve the macro dynamic indicators of the country as they are an accelerator for transformational shifts in the industrial sector of the country and affect GFCF. Consideration of non-produced non-financial assets and non-produced fixed assets is problematic due to difficulties in their evaluation. Therefore, the current condition and development of national wealth indicators are estimated by their basic component—fixed assets. This is the reason for the particular interest in determining the role of infrastructure construction in influencing GFCF assessment.

The construction industry is associated with both the direct and indirect effects of the expenses and revenues of the government budget. For example, this relationship may be indirect in the case of environmental impact (Tardieua et al. 2015). The construction of roads, and especially highways, leads to rapid development of a region, structural change in investments, and alterations in the distribution of funding sources (Xu et al. 2015). Transport infrastructure building projects are frequently connected with significant alterations in land use assets, long-term investments, and financial resources. The construction industry affects sustainability and performance factors in various sectors of the economy and contributes to the implementation of strategies for the sustainable development of countries and regions (Amiril et al. 2014). In periods of economic crisis, transport infrastructure development becomes a vital factor in overcoming stagnation. The relativity between the increase in infrastructure construction volume, real estate construction, and GDP has been widely substantiated (The World Bank 1984; Ozkana et al. 2012; Liu et al. 2016; Okoye et al. 2017), which causes changes in the sustainability of economic growth indicators.

Fluctuations in the economy revolve around certain tendencies in which development is determined by the interaction between revenue and investment size. Construction is a part of the aggregate demand in the form of the requirement for infrastructure establishment; it identifies movement in the short-term and increases the national reserve of productive assets. In addition, the construction sector plays a central role in determining long-term economic growth (Barro 1991; Agbloyor et al. 2014). A variety of multiplicative effects occur between the construction sector and other sectors of the economy (Lopes 1998; Park 1989).

This brief review highlights that construction is viewed as having a causal relationship with changes in GDP, GNP, and changes in the development of adjacent sectors of the economy. Construction is considered and evaluated only as a share of GFCF. We concluded that a high degree of long-run co-movements is present between construction with GFCF and the influence of construction on GFCF.

## 3. Empirical Methodology

The motive behind our article preparation was the absence of analytical research between construction and GFCF in Russia, and little research has been conducted in other countries. Even the studies that are available (Bazilian et al. 2011) have estimated GFCF as a time series to obtain insight into the investment trends in coal, oil, and gas supply, more so than in construction. The evaluations of the share of construction in GFCF is considered as a unit of construction purchasing power parities (PPPs) for marketing research (Gruneberg and Folwell 2013). Since the cause-and-effect relationships between construction and GFCF in the literature are extremely insufficient and only estimated a share (part) of construction in GFCF, we propose investigating construction and GFCF as a discussion of the properties of time-series data by comparing the common data and the testing methods for time-series data analysis using the following parameters: The gross fixed capital formation index (GFCF), the cost index (SUP), the construction work index (RC), and the investment index in fixed capital (IC).

In this case, it is better to use the autoregressive distributed lags (ARDL) model, which indicates that the present value of any variable is determined by its past value and some adjustment factors. An autoregressive distributed lag (ARDL) model is an ordinary least square (OLS)-based model that is applicable for non-stationary time series (Pesaran and Yongcheol 1999; Pesaran et al. 2001).

Time-series data often possess unique features such as clear trends, a high degree of persistence on shocks, and higher volatility over time. The ARDL model is written as:

$$y_t = a_0 + \sum_{i=1}^{p} a_i y_{t-i} + \sum_{j=0}^{q} b_j x_{t-j} + \varepsilon_t \tag{1}$$

where $Y_t$ is a dependent variable $Y$ at period $t$, $X$ is the independent variable, $a$ and $b$ are the parameters with lag indication, and $\varepsilon_t$ is the unexplained part (gap) of the actual data and fitted line by the regression equation, termed as the error. For a one-off unit change in $x$, there is an impact on $y$; this impact is captured by $b_0$; $b_1$ is the impact on y after one period, $b_2$ is the impact after 2 periods, and so on. The final impact on y is $b_k$. If all $t$ coefficients are collected $\{b_0, b_1, b_2, \dots, b_k\}$, they are called the impulse response function of the mapping of $x_t$ to $y_t$. The above model is the lagged model accounting for the changes in $\{ b_0, b_1, b_2, \dots, b_k \}$ on $x$ for lagged period $t$. On the y-axis, the y-dependent may also respond to exogeneous factor; thus, the ARDL may accommodate for both $x$ and $y$.

The formula for calculating the autocorrelation coefficient in the model is written as:

$$\overline{y}_1 = \frac{1}{n-1} \sum_{t=2}^{n} y_t, \overline{y}_2 = \frac{1}{n-1} \sum_{t=2}^{n} y_{t-1} \tag{2}$$

where:

$$\overline{y}_1 = \frac{1}{n-1} \sum_{t=2}^{n} y_t, \overline{y}_2 = \frac{1}{n-1} \sum_{t=2}^{n} y_{t-1}. \tag{3}$$

The vector model of the autoregression is built on the stationary time series. The vector error correction model (VECM) is built in case of a non-stationary time series (AR (1)), which are cointegrated. An order of the model is determined by the order of a lag; the VAR includes two variables with lag 1 and the number of model equations is equal to the number of variables. For $k$ variables and $p$ number of lags, the autoregressive model (VAR(p)) for the matrix model is:

$$X_t = \alpha + A^{[1]} X_{t-1} + \dots A^{[p]} X_{t-p} + \varepsilon_t. \tag{4}$$

Shin's method is frequently applied for assessment of similar models (Shin et al. 2014). Shin developed a non-linear ARDL approach for the research of potential asymmetric effects in both the long- and short-term. In general, the non-linear ARDL approach is based on the linear ARDL approach, which was developed by Pesaran et al. (2001).

First, to test the causal relationship, the Granger test was conducted with the time series of several indexes: The GFCF, SUP, RC, and IC. The Granger test consistently tests two null hypotheses: "X does not Granger-cause Y" and "Y does not Granger-cause X". In this paper, Y is the GFCF, and X is the SUP, the RC, and the IC.

As such studies have not yet been undertaken in Russia, it was necessary to first analyze the quarterly and annual data of the Federal State Statistics of Russia so that variables for the model could be correctly selected. As a result, the process required three stages. The first stage was the identification of causal relationships in the dynamics between the construction market, economic growth, and GFCF in the period from 2000 to 2016. In addition, critical periods of 2007–2009 and 2013–2016 were determined, then the observable data were determined (the variables GFCF, SUP, RC, IC). The second stage involved testing the hypothesis about the form of the model (simple or multiple, linear, or non-linear). The Granger test and the ADF test (assessment of VAD) were conducted to detect exogeneity, stationarity, and cointegration of the time series. The third stage involved evaluating the accuracy of the regressive analysis, time-series data autocorrelation testing, and assessment of the results of ARDL modeling (from 2000 to 2016) with the critical periods (2007–2009; 2013–2016) and comparing the results and the hypothesis of the role of the construction sector in GFCF.

## 4. Results

We implemented the non-linear ARDL approach by estimating Equation (1), which allowed us to determine if there was a long-term relationship between GFCF, SUP, RC, and IC. The techniques applied in the time-series analysis were primarily concerned with the stationarity of the data. The index of supply (SUP) testing on stationarity indicated that Xvar and ADF were not stationary, which was why the SUP index was excluded from the research in the second stage. The statistical procedure of the stationarity of a series included the results of the Granger test (Granger 1969; Cees and Panchenko 2006), the Dickey–Fuller test (Dickey and Fuller 1981), and vector autoregressive (VAR) models (Pesaran et al. 2001) to determine the stationarity, cointegration, and strong connection of the GFCF, RC (construction), and IC (fixed capital investment) variables. If cointegration exists, a regression analysis can reveal the pattern of relationships among the variables of interest, such as GFCF and RC. We concluded that the GFCF and RC variables were linked to form a long-term equilibrium relationship. Finally, we estimated the construction contribution (RC) in the GFCF.

*4.1. Data Issues*

The aim of our empirical research was to identify the causality between gross fixed capital formation and the construction sector. Quarterly and annual data were represented by the Federal State Statistics of Russia in the form of indexes and in the form of monetary quantitative data. The analysis included several indexes: The GFCF, SUP, RC, and IC. These indexes have been deflated with inflation.

One of the major hypotheses in this study was the identification of a causal relationship of the construction industry in Russia as a variable that could (or could not) influence the dynamic of the GFCF. By considering the qualitative nature of the causal model and the complexity of the system, the systematic approach requires both the visualization of its elements and the statistical assessment of the relationship between them. The technique for generating the necessary information about the relationships between the key variables requires both qualitative and quantitative assessment. Therefore, an evaluation of the construction industry market was required. This market is affected by many factors such as the level of administrative barriers to innovation, particularly government regulation and government funding.

The construction sector in Russia has been experiencing a difficult period. The net volume of the construction sector decreased by 13% in the period between 2000 and 2016 years, while, in monetary units, the volume increased from 503 billion rubles in 2000 to 7546 billion rubles in 2016. Positive and stable growth was observed in the period before 2007; the situation changed in 2009 and 2010, and the volume of work performed by the economic activity "Construction" decreased as shown in Figure 1a–c.

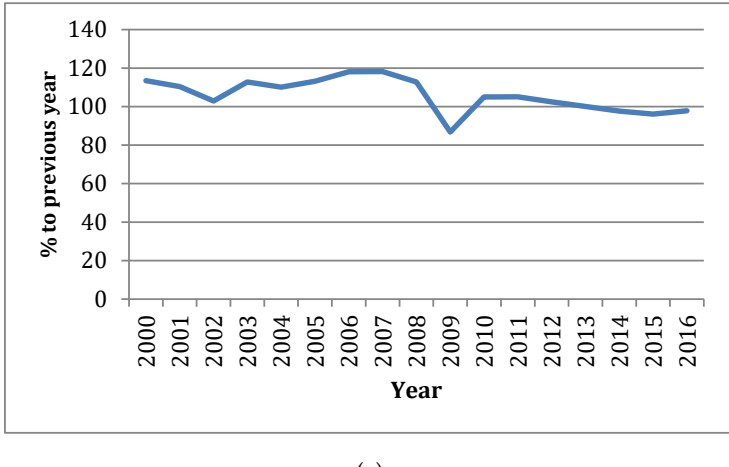

(a)

**Figure 1.** *Cont.*

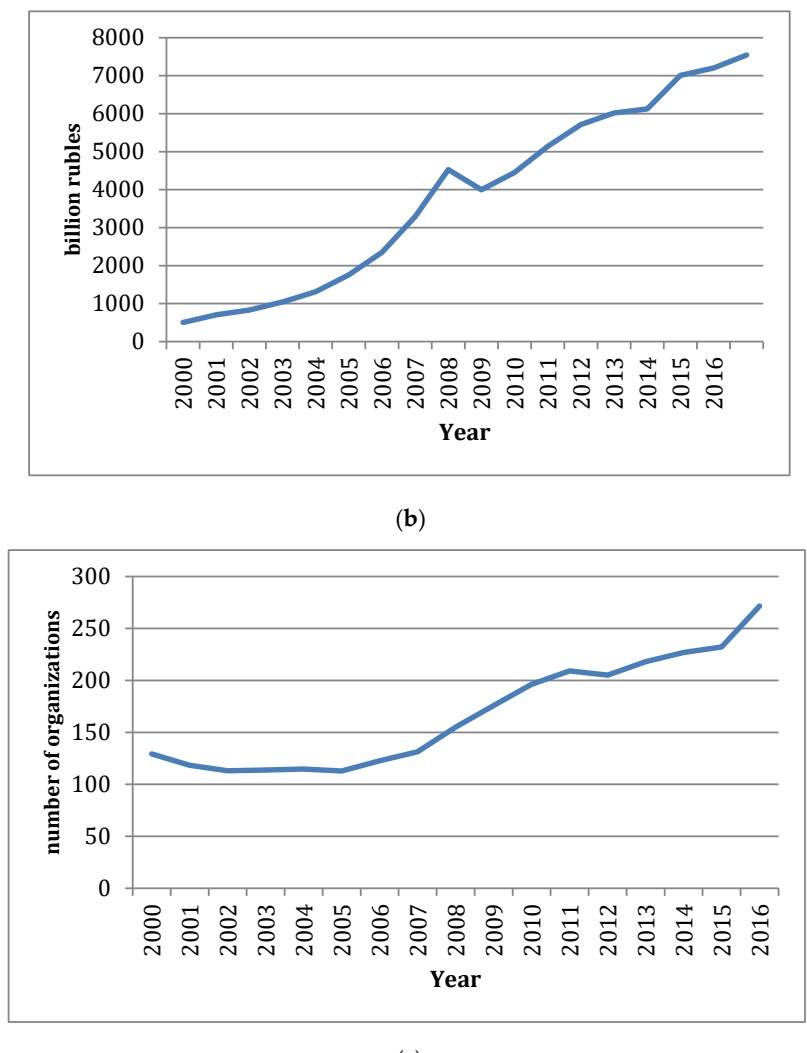

**Figure 1.** Construction in Russia from 2000 to 2016. (**a**) Volume of work performed by the economic activity "Construction" (percentage of corresponding period of previous year); (**b**) volume of work performed by the economic activity "Construction", billion rubles; and (**c**) number of active construction organizations, thousand pieces.

The negative factor is the insignificant demand on services offered by building companies from major investors such as the government, population, and the private corporate sector. The profitability of the construction sector been maintained at the low level of 3.7%; however, the average figure across Russia is 8.1%. Contractors do not have the option to affect the situation, unless they prefer to drop the prices in the industry. Despite this, during the period between 2000 and 2016, the number of operating construction companies increased by 207%. However, low profitability, weak demand, and lack of investments may result in the bankruptcy of companies and further decline in profitability. The major reasons for limitations on the demand for services in the construction sector are both the chronic decrease in income of the citizens and weak investing activity of the corporate sector in terms of modernizing its production with the erection of new buildings and structures. According to the Central Bank of the Russian Federation, the construction sector has the highest level of bad loans. At the beginning of 2017, the percentage of toxic loans in the fourth (problematic loans with high level of risk) and fifth categories (bad loans) in the construction sector was 27.5%. The physical volume of GDP in the first quarter of 2017 was lower than at the beginning of 2012, as production has been stagnating for a long time. This situation is partially explained by the negative external factors: The

rapid drop in oil prices in 2014 and the imposed external sanctions. However, in general, Russian economic problems have internal causes. Economic problems arose in 2013 when the growth of GDP was 1.3%, which has not increased since then. This situation is aggravated by the appearance of new, yet reasonably expected, challenges. Russia has entered a long-term period in the deterioration of demographic indicators: From 2019 to 2025, the size of the labor force will decrease by 0.5%–0.7% per year, which will presumably cause the additional slowdown of economic growth in the amount of 0.5 units. (Ivanova et al. 2017).

Most of the expenses (99.5% on average) of extrabudgetary funds in the expenditure side of the expanded budget are allocated to cultural and social fields (mainly to expenses under the "social policy" item). Before the crisis in 2007, budgetary policy had a pro-cyclical characteristic: With the growth in economy and increase in oil prices, budget expenses expanded. This was especially observed in 2007, when expenses of the expanded budget (in percentage of GDP) increased by more than three percentage points. (Kudrin and Sokolov 2017). Between 2000 and 2010, more than a half (non-percentage) of expanded budget expenses were spent on "culture and social field" units. In the crisis year of 2009, social field expenses noticeably increased from 17.7% of GDP to 21.9% of GDP, partially due to the increase in the pension. During the first three quarters of 2010, expenses associated with social needs were significantly higher (20.7% of GDP), while expenses on units of "national economy, housing, and environment" decreased from 9.8% of GDP to 6.1% of GDP. Without considerable structural reforms, the Russian economy cannot overcome this stagnation. According to the main scenario provided by the Ministry of Economic Development of Russia, the growth in GDP in Russia from 2019 to 2020 was established as being 1.6% on average; the International Monetary Fund (IMF) expects growth rates from 2020 to 2026 of 1.5% per year. Fundamental problems have arisen in the budget system in Russia because the revenue from oil and gas decreased in 2014. Production has decreased, which has led to a reduction in tax revenue. From 2014 to 2016, the budget was planned in expectation of a growth in GDP level in the following three years of 9.7%, while production decreased by 2.3%. In comparison with 2013, the real revenues of the budget system in 2016 were lower by 15% than the Russian government had expected. The problems in revenue budget arose before 2014 (Goryunov et al. 2013) and prior to the drop in the price of oil. Goryunov et al. (2013) suggested that the predicted long-term budget revenue has been exceeded by expected long-term budget expenses, which has resulted in a budget gap 8.4% the size of GDP. Goryunov et al. (2015) claimed that the budget gap has extended to 13.6% of GDP, which was caused by the decline in oil prices in 2014. In other words, the reduction in oil prices aggravated the situation both in the economy and the budget imbalance. For the long-term, a 27.8% budget gap will lead to the growth of expenses to 41.5% of GDP (Kudrin and Sokolov 2017). Therefore, we do not exclude the possibility that Russian debt will increase, and that Russia will experience a debt crisis.

*4.2. Empirical Results*

4.2.1. Results of Parameter Evaluation

For this model, variables that have mostly absorbed the current events were chosen: The index of gross fixed capital formation (GFCF), the index of supply of interindustry balance (SUP), the index of volume of construction (RC), and the index of fixed capital investment (IC). The period of the study (from 2000 to 2016) was characterized by two crisis periods: The first from 2007 to 2009, and the second from 2013 to 2016. The last crisis was so significant that it produced the current stagnation in 2019. The construction industry, as a sector that generates products with a long life cycle, is especially sensitive to such economic shocks (Figure 2).

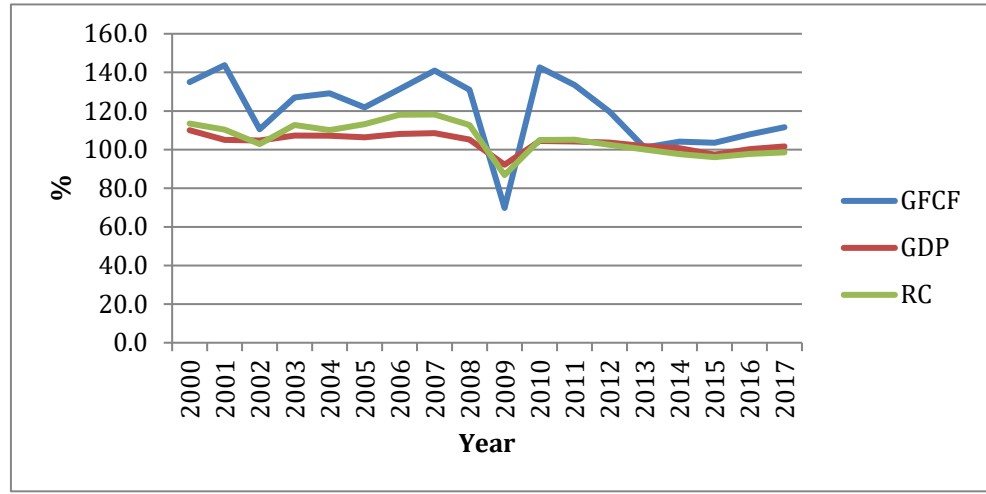

**Figure 2.** Gross fixed capital formation (GFCF), gross domestic product (GDP), and gross value construction (RC) in Russia, %.

### 4.2.2. Results of Unit Root Tests

Experimentally through the Granger Test, the causal relationships between GFCF and a range of indexes (RC, IC, and SUP) were determined. The key variable was the GFCF. The correlation with GFCF in Table 1 demonstrates that GFCF has a strong dependence on SUP and RC, and a much weaker dependence on IC. The correlation of the causal relationship in Table 2 demonstrates that SUP expenses strongly depend on GFCF, in contrast to RC, which does not have such dependence. IC weakly depends on GFCF.

**Table 1.** Results of the Granger test: The effect of the dependence of GFCF on SUP, RC, IC.

| lag | ΔGFCF = f(SUP) | ΔGFCF = f(RC) | ΔGFCF = f(IC) |
|-----|-----|-----|-----|
| 7 | 0.985045812 | 0.932874266 | 0.499845848 |
| 6 | 0.987725055 | 0.905850114 | 0.569685548 |
| 5 | 0.965343381 | 0.85755161 | 0.727798006 |
| 4 | 0.881776951 | 0.840451859 | 0.804727944 |
| 3 | 0.832232471 | 0.821348508 | 0.817885899 |
| 2 | 0.859457287 | 0.874365681 | 0.765398018 |
| 1 | 0.905539034 | 0.893287842 | 0.814934908 |

**Table 2.** Results of the Granger test: The effect of the dependence of SUP, RC, IC on GFCF.

| lag | ΔSUP = f(GFCF) | ΔRC = f(GFCF) | ΔIC = f(GFCF) |
|-----|-----|-----|-----|
| 7 | 0.913892076 | 0.448950278 | −0.616410269 |
| 6 | 0.966367446 | 0.679542729 | 0.14622655 |
| 5 | 0.962124147 | 0.738654516 | 0.454208722 |
| 4 | 0.915996188 | 0.769489618 | 0.412950781 |
| 3 | 0.908163404 | 0.792722071 | 0.466605836 |
| 2 | 0.918128928 | 0.721287279 | 0.556031222 |
| 1 | 0.933430426 | 0.787836355 | 0.711410163 |

The model of the dynamics of several time series (VAR) (Pesaran et al. 2001) demonstrated dependence on the previous values of the same time series. The augmented Dickey–Fuller test (ADF) (Dickey and Fuller 1981) for the case of deviation of the null hypothesis of the presence of a unit root (where *t* statistics are lower than the marginal value) indicated that the marginal value for ADF has its own distribution, which was −2.89. The RC index was 1.862 at the 1% level and 2.668 and the 5%,

thus, we concluded that the null hypothesis was rejected, and the RC and GFCF data are cointegrated (Table 3).

**Table 3.** Assessment of a time series (augmented Dickey–Fuller (ADF) test).

| Variable | Xvar | | Level/-Distribution | |
|---|---|---|---|---|
| | **1%** | **5%** | **1%** | **5%** |
| inGFCF | 0.960 | 0.960 | 3.906 | 2.700 |
| inSUP | 0.850 | 0.940 | 3.620 | 4.490 |
| inRC | 0.972 | 0.920 | 1.862 | 2.668 |
| inIC | 0.902 | 0.902 | 3.760 | 3.239 |

The index of supply (SUP) testing on stationarity indicated that $X_{var}$ (0.850–0.940) and ADF (4.490 at 5%) were not cointegrated, which why the SUP index was excluded from our research.

### 4.2.3. Results of the ARDL Models

We assumed that all exogenous variables were included in the ARDL model (p, r; s), where p is the number of lags of the dependent variable, r is the number of lags of independent variables, and s is the number of independent variables. In the prediction of values with ARDL, limitations on these models were included in our consideration (Pesaran et al. 2001). The application of the ARDL model is only possible when the time series is stationary, which is calculated using the Dickey–Fuller test (Dickey and Fuller 1981). Sequential correlation test results indicated that the error terms were normally distributed, serially independent, and homoscedastic at the 1%, 5%, and 10% levels of significance. Finally, the results of the CUSUM square tests for the dependence of the variables displayed the greater importance of the dependency coefficients (0.7–0.9) for almost all parameters. Therefore, the coefficients of the models were dynamically stable: In the series of dynamics, there is a tendency to 1. Thus, we concluded that the selected ARDL models were correct and reliable. The results are summarized in Table 4.

**Table 4.** The autoregressive distributed-lagged (ARDL) estimation results. GFCF, RC, and IC responses (2000–2016), annual data.

| Lag | Dependent Variable | Coefficient | *t*-Statistic | *p*-Value |
|---|---|---|---|---|
| 1 | $F_{GFCF}(GFCF/RC,CI)_{t-1}$ | 0.974 *** | 23.38 | 0.954 |
| | $F_{RC}(RC/GFCF,CI)_{t-1}$ | 0.9858 *** | 32.01 | 0.986 |
| | $F_{IC}(IC/GFCF,RC)_{t-1}$ | 0.9236 *** | 12.94 | 0.924 |
| 2 | $F_{GFCF}(GFCF/RC,CI)_{t-2}$ | 0.947 *** | 15.43 | 0.947 |
| | $F_{RC}(RC/GFCF,CI)_{t-2}$ | 0.9861 *** | 21.4 | 0.971 |
| | $F_{IC}(IC/GFCF,RC)_{t-2}$ | 0.8769 *** | 12.94 | 0.924 |
| 3 | $F_{GFCF}(GFCF/RC,CI)_{t-3}$ | 0.974 *** | 23.38 | 0.937 |
| | $F_{RC}(RC/GFCF,CI)_{t-3}$ | 0.9627 *** | 17.97 | 0.963 |
| | $F_{IC}(IC/GFCF,RC)_{t-3}$ | 0.8383 *** | 7.52 | 0.868 |
| 4 | $F_{GFCF}(GFCF/RC,CI)_{t-4}$ | 0.947 *** | 13.39 | 0.941 |
| | $F_{RC}(RC/GFCF,CI)_{t-4}$ | 0.9548 *** | 15.56 | 0.955 |
| | $F_{IC}(IC/GFCF,RC)_{t-4}$ | 0.7885 *** | 5.94 | 0.924 |

Notes: $F_{GFCF}$ (GFCF/RC,CI)$_{t-1}$ is the dependent parameter functions with lag indication (Pesaran et al. 2001); *, ** and *** denote the statistical significance at 10%, 5% and 1% significance levels, respectively (Dickey and Fuller 1981).

The assessment of the non-linear correlation of variables demonstrated that in 2000–2016, there were at least two crisis (shocking) periods: 2006–2010 and 2013–2016. Therefore, shocking periods have to be observed in quarterly detail. Table 4 shows that a high level of dependence was observed in the $F_{RC}(RC/GFCF,CI)$. If the $F_{RC}(RC/GFCF,CI)$ dynamics of the tendency are strong (F(RC)$_{t-1}$ = 0.6345 → 1),

$(F(RC)_{t-4} = 0.7943 \rightarrow 1)$, then independent of $F_{GFCF}(GFCF/RC,CI)$ and $F_{IC}(IC/GFCF,RC)$, the dynamics of dependence are not observed $(F \rightarrow 0)$. The results are provided in Table 5.

**Table 5.** The ARDL estimation results. GFCF, RC, IC responses to external shocks (2006–2010), quarterly data.

| Dependent Variable | Coefficient | | *t*-Statistic | *p*-Value |
|---|---|---|---|---|
| $F_{GFCF}(GFCF/RC,CI)_{t-1}$ | 0.3988 | ** | 0.62 | 0.222 |
| $F_{GFCF}(GFCF/RC,CI)_{t-2}$ | 0.3129 | ** | – | – |
| $F_{GFCF}(GFCF/RC,CI)_{t-3}$ | −0.3972 | ** | – | – |
| $F_{GFCF}(GFCF/RC,CI)_{t-4}$ | −0.2222 | ** | – | – |
| $F_{RC}(RC/GFCF,CI)_{t-1}$ | 0.6345 | *** | 4.29 | 0.794 |
| $F_{RC}(RC/GFCF,CI)_{t-2}$ | 0.3193 | ** | – | – |
| $F_{RC}(RC/GFCF,CI)_{t-3}$ | 0.5055 | *** | – | – |
| $F_{RC}(RC/GFCF,CI)_{t-4}$ | 0.7943 | *** | – | – |
| $F_{IC}(IC/GFCF,RC)_{t-1}$ | 0.3065 | ** | 1.03 | 0.341 |
| $F_{IC}(IC/GFCF,RC)_{t-2}$ | −0.1906 | ** | – | – |
| $F_{IC}(IC/GFCF,RC)_{t-3}$ | 0.06296 | * | – | – |
| $F_{IC}(IC/GFCF,RC)_{t-4}$ | 0.3408** | ** | – | – |

Notes: $F_{GFCF}(GFCF/RC,CI)_{t-1}$ is the dependent parameter functions with lag indication (Pesaran et al. 2001); *, ** and *** denote the statistical significance at 10%, 5% and 1% significance levels, respectively (Dickey and Fuller 1981).

Table 6 shows that in the period of 2013–2016, the strong dependence of $F_{RC}$ (RC/GFCF, CI) remained. Periodic oscillations in $F_{GFCF}(GFCF/RC,CI)$ have a period of four.

**Table 6.** The ARDL estimation results. GFCF, RC, and IC responses to external shocks (2013–2016), quarterly data.

| Dependent Variable | Coefficient | | *t*-Statistic | *p*-Value |
|---|---|---|---|---|
| $F_{GFCF}(GFCF/RC,CI)_{t-1}$ | 0.4817 | ** | 1.56 | 0.466 |
| $F_{GFCF}(GFCF/RC,CI)_{t-2}$ | −0.01816 | * | – | – |
| $F_{GFCF}(GFCF/RC,CI)_{t-3}$ | −0.354 | * | – | – |
| $F_{GFCF}(GFCF/RC,CI)_{t-4}$ | −0.4656 | * | – | – |
| $F_{RC}(RC/GFCF,CI)_{t-1}$ | 0.7546 | *** | 3.3 | 0.717 |
| $F_{RC}(RC/GFCF,CI)_{t-2}$ | 0.6643 | *** | – | – |
| $F_{RC}(RC/GFCF,CI)_{t-3}$ | 0.6594 | ** | – | – |
| $F_{RC}(RC/GFCF,CI)_{t-4}$ | 0.7169 | *** | – | – |
| $F_{IC}(IC/GFCF,RC)_{t-1}$ | 0.3065 | ** | 1.03 | 0.341 |
| $F_{IC}(IC/GFCF,RC)_{t-2}$ | −0.1906 | * | – | – |
| $F_{IC}(IC/GFCF,RC)_{t-3}$ | 0.06296 | * | – | – |
| $F_{IC}(IC/GFCF,RC)_{t-4}$ | 0.3408 | ** | – | – |

Notes: $F_{GFCF}(GFCF/RC,CI)_{t-1}$ is the parameter-dependent functions with lag indication (Pesaran et al. 2001); *, ** and *** denote the statistical significance at 10%, 5% and 1% significance levels, respectively (Dickey and Fuller 1981).

The multiplicative effects of the influence of the construction sector RC on GFCF are defined as the sum of the gross value added. The increase of gross value added in GFCF can be assessed as the growth of the output in construction sector:

$$\Delta RC_{GFCF} = \sum_{i=0}^{t} \Delta GFCF = \sum_{i=0}^{t} \Delta X_i * \frac{GFCF_i}{X_i} \tag{5}$$

where $\Delta RC_{GFCF}$ is the index of construction, $X_i$ is the gross construction output, $\Delta X_i$ is the increase in the gross construction output, and $\Delta GFCF$ is the increase in the gross value added.

Based on the regression analysis evaluation of the multiplicative effects of the influence of construction sector RC on GFCF, the hypothesis for the general aggregation of the correlation among all possible values of GFCF and RC was confirmed. The regressive equation is written as

y = 0.944x + 0.0826. With the OLS approximation, the assessment of regressive equation parameters, which are characteristic of a specific statistical observation of GFCF and RC, was obtained. The stationarity of the time series was calculated with the application of the Dickey–Fuller test (Dickey and Fuller 1981). To assess the statistical significance of the regression and correlation coefficients, the critical values of the F-test were calculated (Pesaran et al. 2001). Heteroscedasticity was verified using the Goldfeld–Quandt test (Goldfeld and Quandt 1965).

We found that, in the situation under study, 61.92% of the total variability of GFCF could be explained by the change in RC. The regression coefficient *b* = 0.944 indicates an average change in the GFCF effective index with an increase or decrease in the value of the RC index per unit of measurement. When the RC value increased by 1 point, the GFCF increased by 0.944 points on average. The data of the ARDL model in Equation (1) and regression analysis evaluation of the multiplicative effects of the influence of construction sector RC on GFCF using Equation (1) are displayed in Table 7.

**Table 7.** Results of the ARDL model of the relationship between GFCF and RC.

| Indicator | Value |
| --- | --- |
| Sum of squares | 0.363 |
| Number of degrees of freedom | 18–1 |
| F-criterion | 26.02 |
| Goldfeld–Quandt test | 1.24 |
| Coefficient of determination | 0.6192 |
| Coefficient of elasticity | 0.93 |
| Mean approximation error | 7.35 |

Notes: Sum of squares (Markowski and Markowski 1990); F-criterion is the ADF test (Pesaran et al. 2001); checks for homoscedasticity (Goldfeld and Quandt 1965); coefficient of elasticity is the indicator of alteration of parameters to changes; coefficient of determination is the GFCF variability; mean approximation error is the difference between the exact value and the approximate value.

Diagnostic tests of the ARDL model (Tables 4–6) tell us about the robustness of the estimated coefficients. In time-series analysis (Table 7) of the ARDL model, we can understand the behavior of the variables, their interactions, and integrations over time. We did not observe a strong upward or downward movement over time with no tendency to revert to a fixed mean GFCF and RC: The sum of squares (dispersion on 1 degree of freedom) was 0.363; and the coefficient of determination was 0.6192 (at number of degrees of freedom was 18–1). The F-criterion was 26.02 ($F_{tabl.}$ = 4.49). Since the actual value of F > $F_{tabl}$, the estimated autoregression was a time series found to be statistically reliable. The elasticity coefficient was 0.93 (93%), indicating that a percentage change will occur in the variable GFCF when the RC variable changes 1%, that is, small changes in RC do modify the GFCF. The mean approximation error was 7.35; the calculated values deviated from the actual by 7.35%. Thus, the diagnostic tests, the ARDL model $F_{GFCF}$(GFCF/RC,CI)$_{t-k}$, and its analysis using linear regression ($\Delta RC_{GFCF}$) confirm the choice of model.

In conclusion, the relationship between the gross fixed capital formation and the volume of construction at the 5% level of significance is displayed in Figure 3.

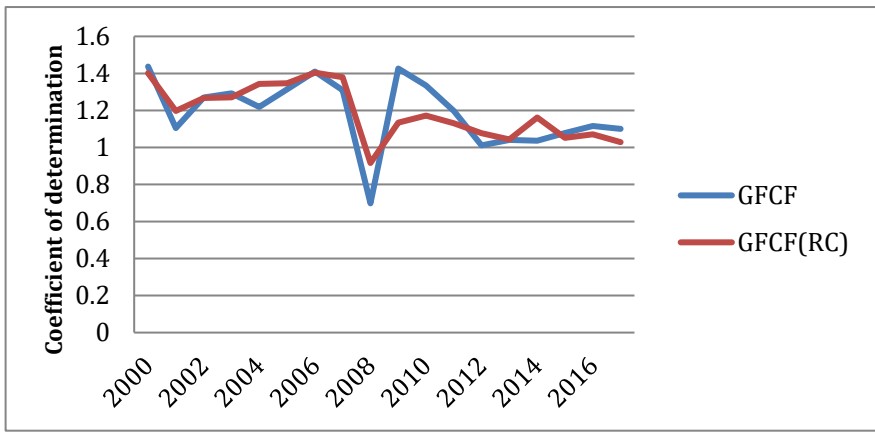

**Figure 3.** Results of the model GFCF (RC).

## 5. Discussion

During the period 2009–2019 revenue in the Russian budget has decreased due to lower oil and gas revenue. The outcomes of the slow economic growth of Russia have to be explained. The Russian government explains it as the reaction to sanctions and the obligatory need to eliminate the outcomes of market failures, provide public goods, and accomplish social objectives. Two crises (2006–2009; 2013–2016) also have to be explained. With significant financial resources, the government funds infrastructure construction and real estate development. The aim of this study was to detect the connections, trends, and the high degree of persistence of the shocks to the construction sector and GFCF through the ARDL model and regression analysis. As discussed above, no studies have focused on the assessment of the role of the construction sector in additional gross fixed capital formation through the ARDL model method in Russia. Therefore, in the first stage of research, statistical data for the period between 2000 and 2016, which were provided by the Federal State Statistics of Russia, were used to establish the cause and effect relationship. Expenses in the development of infrastructure varied from 30% to 32% of the government budget in different years in the period. At the end of the first stage of the study, the most important variables for the model were identified: SUP, RC, IC, and GFCF. In the second stage of the study (the ARDL model method, ADF test, and the Granger test were applied), the SUP index was eliminated due to the lack of cointegration of the time series and due to its exogeneity. All the ARDL models, which are defined in Equations (1)–(4), succeeded through all of the diagnostic tests. The results demonstrated that there is a long-term relationship between RC and GFCF at the 1%, 5%, and 10% levels of significance. These variables and the internal capital investment in construction have a cointegrated relationship, and oscillation exists in their relationship caused by shocking events (crises). Further investigation demonstrated the presence of negative shocks and the influence asymmetries of IC to GFCF in two periods: 2007–2009 and 2013–2016. The analysis of quarterly data through the ARDL modeling method in these periods showed that only the RC index demonstrated significantly pronounced stationarity in connection with GFCF. Finally, the hypothesis of the significant role of the construction sector in the GFCF of Russia was confirmed with the usage of ARDL modeling and linier regressive model tests, concluding that the volume of work produced by the construction sector (RC) has a noticeable influence on the GFCF. We established that one ruble of expenses in the sector leads to a growth in GFCF by 0.944 rubles on average. The construction sector is a stable parameter of additional value of the GFCF; even in crisis periods, it represents 61.92% of the entire variability of the GFCF.

## 6. Conclusions

Empirical analysis of the literature of the construction sector role (RC) in gross fixed capital formation (GFCF) showed that analytical studies relied heavily on identifying construction only

as a share of RC in GFCF (Bazilian et al. 2011; Gruneberg and Folwell 2013; Lopes 1998; 1987; Amiril et al. 2014; Ozkana et al. 2012; Okoye et al. 2017), considering the construction sector as a factor of economic growth. Unlike these studies, we turned to the GFCF and RC statistics as a sequence of observations of the defined variables at a uniform interval over a period of time in successive order to identify the relationship between GFCF and RC. This allowed us to consider RC not only as a share of GFCF, but also determine the stationarity, trends, cycles, seasonality, and structural breaks between GFCF, RC, IC, and GFCF,RC. The results showed that the economic times-series data GFCF(RC,IC) possessed unique features such as a clear trend and a high degree of persistence after shocks. The Granger test, Dickey–Fuller test, autoregressive (VAR) models, and ARDL model (Tables 3–6) determined the stationarity, cointegration, and strong connection of all of the variables (GFCF, RC, and IC). We concluded that the GFCF and RC variables are linked to form a long-term equilibrium relationship. The stability of communication between GFCF and RC remained in both periods of growth and in periods of crisis (2006–2010, 2013–2016). We did not observe a strong upward or downward movement over time and there was no tendency to revert to a fixed mean GFCF and RC (dispersion of freedom was 0.363). Finally, we estimated the contribution of construction (RC) to the GFCF, where small changes in RC did modify the GFCF in 93% cases. We established that one ruble of expenses in the sector leads to the growth of GFCF by 0.944 rubles on average. The regression analysis of the variables GFCF and RC, which we conducted after analyzing the time series GFCF, RC, introduced additional reliability in checking stationarity, trends, cycles. We can argue that this estimate is robust and reliable compared to the estimates given by other methods and applicable to other areas. This assessment might be an additional factor for consideration in some cases when selecting government investment projects and formatting government policy on economic development. In the future, a study that applies disaggregated data for different sectors of the economy and the GFCF could provide results that are more precise.

**Author Contributions:** Both authors contributed equally to this research.

**Funding:** This research received no external funding.

**Conflicts of Interest:** The authors declare no conflict of interest.

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
