# Peer review of "Construction Sector Role in Gross Fixed Capital Formation: Empirical Data from Russia"

_economies, doi:10.3390/economies7020042_

Round 1

Reviewer 1 Report

Introduction

Point 1: Within the first paragraph, there seems to be substantial transition problem from line 30 to 31. "Russia [] One of the..."

Point 2: Write GFCF as Gross Fixed Capital Formation.

Point 3: On page 1, line 33: It is claimed that GFCF is not an indicator of investment. However, most of the macroeconomic literature takes it as an indicator of capital for growth equation. Authors' claim is not satisfactory and seems problematic in terms of empirical investigation. 

Literature Review

Point 1: Literature review should be extended and written analytically. 

Empirical Methodology

Point 1: Method is not clearly explained. 

Point 2: The motivation for the investment and GFCF relationship is not clear.

Point 3: Moreover research question is not analytical and does not have a theoretical background. 

Conclusion

Point 1: Not satisfactory and written very shortly. 

Others

Point 1: The reference on page 2, line 89 seems wrong Ozkona or Ozkan? or "ARDL" modelling method" on page 13, line 405.

Point 2: There typo mistakes such as on line 185. "non-linear" should be "no-linear".

Point 3: Language is problematic. 

I suggest to reject paper for publication since it does not satisfy the quality standards of the journal based on my comments above. 

Author Response

Dear Reviewer 1,

Thank you for your email enclosing the comments. We

have carefully reviewed the comments and have revised the manuscript accordingly. Our responses are given in a point-by-point manner in the attachment (Word).

We used the services English editing. 

The article is completely corrected taking into account your comments and editors' notes. We are ready to send a new version of the article.

Reviewer 2 Report

The paper analysis the role of the construction sector versus the gross fixed capital formation.  Both the introduction and the literature review are well written and provide comprehensive aspects related to the gross domestic product, the events in Russia and the researches conducted in this field from all around the world. Before acceptance, I believe that the following issues should be considered:

-  please highlight the variables introduced in equation 1;

- please add some explanations on how you intend to determine the results in section 4 just before section 4.1.;

- please add legend to figure 1 and some values within the figure (if it is not possible, simple add a summary table with the data you have used);

- what is the range for "i" in equation 5?

- please move the citation on row 360 in row 359;

- please add the names of the axes in Figure 3 and use the same template as in the previous figures. Please all point the values of the data used;

- please extend the conclusions and refer to your work in comparison with some of the paper presented in the literature review section. Can you point any differences?

Thank you!

Author Response

Dear Reviewer 2,

Thank you for your email enclosing the comments. We

have carefully reviewed the comments and have revised the manuscript accordingly. Our responses are given in a point-by-point manner in the attachment (Word).

We used the services English editing. 

The article is completely corrected taking into account your comments and editors' notes. We are ready to send a new version of the article.

Reviewer 3 Report

1. The results are clearly presented, but I suggest a more strucured presentation. For instance, I suggest to insert a table with the main results from the analysis and then to analyze the results. 2. The conclusions should be related to the analysis and they should be presented in detail.

Author Response

Dear Reviewer 3,

Thank you for your email enclosing the comments. We

have carefully reviewed the comments and have revised the manuscript accordingly. Our responses are given in a point-by-point manner in the attachment (Word).

We used the services English editing. 

The article is completely corrected taking into account your comments and editors' notes. We are ready to send a new version of the article.

Round 2

Reviewer 1 Report

Dear Editor,

Please find my final comments below.

Point 1: Authors improved the paper substantially and made a good effort to increase its quality.

Point 2: I also suggest authors to read Sahin (2018) for a comparison of other methods with Autoregressive Distributed Lag Model (ARDL).

Point 3: Sahin et al. (2019) also benefit from Nonlinear Autoregressive Distributed Lag Model (NARDL) that summarize the methodology clearly and applies to a topic related to monetary economics. 

Point 4: Equations such as (2) and (3) have big formats compared to text fonts.

References

Sahin, Afsin (2018). “Staying Vigilant of Uncertainty to Velocity of Money: An Application for Oil-Producing Countries”. OPEC Energy Review, 42(2): 170-195.

Sahin Afsin, Berument, Hakan (2019). “Asymmetric Effects of Central Bank Funding on Commercial Bank Sector Behavior”. Economic Research-Ekonomska Istraživanja, 32(1): 128-147.

Author Response

Dear Reviewer,

Thank you for your email enclosing the comments. We have carefully reviewed the comments and have revised the manuscript accordingly. Our responses are given in a point-by-point manner in the attachment (Word).

We are especially grateful for the suggestions regarding in the consideration of methodology and the articles of Sahin.

We believe that suggestions have been very helpful in improving the manuscript, both in the introduction and methodology and especially in the conclusions.
